# ExoViP: Step-by-step Verification and Exploration with Exoskeleton Modules for Compositional Visual Reasoning

**Yuxuan Wang**[1]    **Alan Yuille**[2]    **Zhuowan Li**[2✉]    **Zilong Zheng**[1✉]

[1] State Key Laboratory of General Artificial Intelligence, BIGAI, Beijing, China

[2] Johns Hopkins University, Baltimore, Maryland, USA

wangyuxuan1@bigai.ai    {ayuille1,zli110}@jhu.edu    zlzheng@bigai.ai

## Abstract

Compositional visual reasoning methods, which translate a complex query into a structured composition of feasible visual tasks, have exhibited a strong potential in complicated multi-modal tasks. Empowered by recent advances in large language models (LLMs), this multi-modal challenge has been brought to a new stage by treating LLMs as few-shot/zero-shot planners, *i.e.*, vision-language (VL) programming (Gupta & Kembhavi, 2023). Such methods, despite their numerous merits, suffer from challenges due to LLM planning mistakes or inaccuracy of visual execution modules, lagging behind the non-compositional models. In this work, we devise a "plug-and-play" method, EXOVIP, to correct errors in both the planning and execution stages through introspective verification. We employ verification modules as "exoskeletons" to enhance current VL programming schemes. Specifically, our proposed verification module utilizes a mixture of three sub-verifiers to validate predictions after each reasoning step, subsequently calibrating the visual module predictions and refining the reasoning trace planned by LLMs. Experimental results on two representative VL programming methods showcase consistent improvements on five compositional reasoning tasks on standard benchmarks. In light of this, we believe that EXOVIP can foster better performance and generalization on open-domain multi-modal challenges.

**Code**   https://github.com/bigai-nlco/ExoViP

# 1   Introduction

Compositional visual reasoning methods, due to their interpretability and generalization over complex vision-language (VL) challenges that demand intricate, multi-step visual reasoning guided by linguistic input, have long been the focus for many researchers. Traditional compositional techniques, exemplified by neural modular networks (Andreas et al., 2015; Hu et al., 2017; Johnson et al., 2017; Hu et al., 2018; Le et al., 2022; Qian et al., 2022), have shown success in breaking down intricate language instructions into manageable visual tasks. However, they tend to falter when confronted with the need for broader generalization across diverse domains. Furthermore, the limitations of these approaches manifest in their inability to enhance the interaction and attention between neural modules through supervision or feedback mechanisms, thereby constraining performance to end-to-end training paradigms. Recent advances in large language models (LLM) (Radford & Narasimhan, 2018; Radford et al., 2019; Brown et al., 2020; OpenAI, 2023; Chowdhery et al., 2022) have led to novel methods that harness LLM as zero-shot or few-shot planners to address visual reasoning tasks, notably vision-language programming (VISPROG) (Gupta & Kembhavi, 2023) and ViperGPT (Dídac et al., 2023). These approaches make use of readily available pretrained vision models and systematically assemble them, guided by the reasoning trace provided by LLMs, resulting in interpretable intermediary outcomes and highly adaptable reasoning capabilities.

---

✉ Corresponding authors.

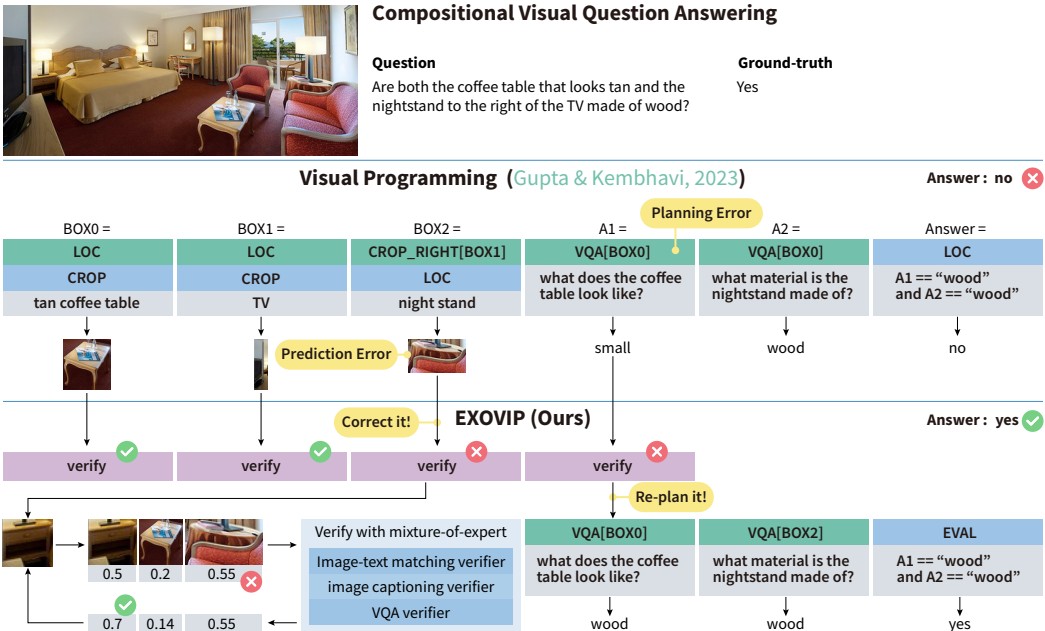

Figure 1: **An overview of EXOVIP.** The prediction after each step is verified by the proposed "exoskeleton" verification modules, which contain a mix of three sub-verifiers. The verified scores help correct the errors in the vision module predictions or refine the reasoning programs planned by LLM.

Despite their merits, current visual programming approaches encounter persistent challenges, often resulting from shortcomings in the planning processes of LLMs or the capabilities of visual modules. More precisely, they often fall short of the performance achieved by non-compositional models; refer to Fig. 1 for exemplar error cases. To investigate these limitations, a manual examination of 100 randomly selected failure cases (**Section 4.5.1**) of VISPROG (Gupta & Kembhavi, 2023) on the GQA dataset (Hudson & Manning, 2019) for visual question answering was conducted. The analysis revealed two primary failure categories: Firstly, approximately 30% of these failures were attributed to **planning errors**, where the LLM failed to parse the language query into programs correctly, preventing the formulation of a solvable program. Secondly, over 40% of the failures were attributed to **module errors**, as the visual modules were incapable of executing the program accurately. The remaining failure cases (less than 30%) stemmed from issues like synonym usage (*e.g.*, "woman" *vs.* "lady") or question ambiguity.

Motivated by these failure modes, in this work, we introduce EXOVIP, a "plug-and-play" method that uses "exoskeleton" verification modules to verify the reasoning results step by step, thus correcting the module errors and refining the LLM planning traces. As depicted in Fig. 1, EXOVIP effectively rectifies both types of errors: the verification module contains a mixture of three sub-verifiers, including an image-text matching verifier, an image captioning verifier, and a visual question answering (VQA) verifier. These sub-verifiers meticulously validate the accuracy of the predictions generated by the visual modules, thereby correcting module errors. For refining the planning traces, a reasoning trace tree is constructed based on the verification scores, along with the self-correctness score (Pan et al., 2023) obtained from LLMs. The methodology involves searching through the tree to identify the optimal trace with the highest score.

To demonstrate the effectiveness of EXOVIP, we apply our method to two recent visual programming methods: self-defined programs, *i.e.*, VISPROG (Gupta & Kembhavi, 2023) and Python code programs, *i.e.*, ViperGPT (Dídac et al., 2023). Our experiments encompass six compositional visual reasoning tasks, including compositional image question answering, referring expression understanding, natural language for visual reasoning, visual abstract reasoning, language-guided image editing, and spatial-temporal reasoning. The

experimental results consistently indicate notable improvements in performance for both models. In light of this, we believe EXOVIP can foster better performance on open-world compositional reasoning tasks.

In summary, our main contributions are as follows:

- We introduce "exoskeleton" verification modules tailored for module error and plan error in existing compositional visual reasoning methods, which systematically validate the accuracy of vision module predictions in a step-by-step manner.
- We illustrate the synergistic integration of our proposed verification modules with a tree-based search algorithm, enhanced by the self-correcting capabilities of the LLM. This collaborative design effectively tackles both the module error and plan error. The tree-based search is informed by a verification score, which serves as a measure of confidence in the search process. Concurrently, this verification score is dynamically refined as the search progresses, ensuring a more accurate and reliable verification process.
- We have implemented our methodology within two compositional methods, and the outcome has been a uniform enhancement in performance across six diverse tasks encompassing both image and video modalities. This underscores the efficacy of EXOVIP in augmenting visual reasoning skills.

## 2   Related work

**LLMs in multi-modal tasks.**   LLMs have significantly enhanced multi-modal tasks through their adaptability and extensive knowledge. There are three primary methods for applying LLMs to multi-modal challenges. One approach involves integrating extra parameters into LLMs for multi-modal contexts and then fine-tuning with either a fixed (Tsimpoukelli et al., 2021; Alayrac et al., 2022; Li et al., 2023b; Gao et al., 2023; Li et al., 2023a; Dai et al., 2023; Zhang et al., 2023d) or an adjustable LLM (Hao et al., 2022; Huang et al., 2023; Peng et al., 2023). Another strategy uses LLMs as knowledge experts, combining them with specialists in other fields like vision and speech to tackle diverse tasks (Zeng et al., 2023; Zhang et al., 2023c; Liu et al., 2023b). Our research concentrates on a third method that leverages the LLM's ability to parse complex queries and delegate tasks to expert agents, whether through custom programs (Gupta & Kembhavi, 2023), Python code (Dídac et al., 2023), or dialogue agents (Yang et al., 2023). However, the effectiveness of these approaches is limited by the quality of the planning sequences and visual experts.

**Compositional multi-modal methods.**   At an early stage, neural module networks (NMN) (Andreas et al., 2015; Hu et al., 2017; Johnson et al., 2017; Hu et al., 2018; Le et al., 2022; Qian et al., 2022; Wang et al., 2024) create end-to-end differentiable networks with neural models, but their pre-set modules struggle with open-domain tasks, and the complex embedding and attention mechanisms hinder the creation of intermediate supervision signals. Recently, the presence of LLMs has made it possible to automatically compose various kinds of finetuned neural models (Zeng et al., 2023; Gupta & Kembhavi, 2023; Dídac et al., 2023; Yang et al., 2023; Liu et al., 2023b) or external tools (Parisi et al., 2022; Khot et al., 2023; Schick et al., 2023; Shen et al., 2023; Lu et al., 2023; Qin et al., 2023). These works allow us to diagnose the intermedia rationales of the reasoning process. However, human annotation of these intermedia results can be rather time-consuming. In this work, we make ways to correct errors in the intermedia results without any human intervention.

**Self-correction in LLMs.**   Although LLMs achieve great success in various tasks, there are many errors in LLM-based system (Pan et al., 2023): hallucination (Li et al., 2023c; Zhang et al., 2023b), unfaithful reasoning (Golovneva et al., 2022; Ribeiro et al., 2023; LYU et al., 2023), toxic, biased and harmful contents (Shaikh et al., 2022), flawed code. One way to fix these errors is to use LLMs themselves (Madaan et al., 2023; Shinn et al., 2023; Ye et al., 2023; Yan et al., 2023) to obtain feedback to correct the errors. Incorporating self-correction strategies from LLMs, researchers aim to streamline reasoning in multi-modal systems. IPVR (Chen et al., 2023) employs LLMs for rationale generation and cross-modality validation to ensure consistent inference. IdeaGPT (You et al., 2023) uses an LLM to summarize and iteratively refine the output of visual experts. To overcome the inherent limitations of

LLM self-correction, our approach merges LLM feedback with insights from visual experts to authenticate intermediate results and the reasoning process.

# 3 EXOVIP

To address the aforementioned shortcomings, we propose EXOVIP. This framework adopts exoskeleton verification modules to calibrate the prediction of the execution modules and refine the reasoning path with tree searching. In this section, we will first introduce the preliminaries, including our task definition and visual programming. Then we will show the verification modules, and describe how the verification results are applied to correct the results of execution modules and to search for the reasoning trace. Additionally, we will introduce two mechanisms – negative sampling and post-hoc self-correction to alleviate extra errors introduced by verification modules.

## 3.1 Preliminaries

**Task definition.** Our work focuses on Visual Compositional Reasoning (VCR) tasks. These VCR tasks require reasoning on a series of steps about an image input $I$ and a text input $T$, and predict the output, *e.g.* answer to a given question, edited images given a language instruction, etc.

**Visual programming (VISPROG).** VISPROG (Gupta & Kembhavi, 2023) is a zero-shot VCR model that leverages LLMs and pretrained vision models. It transforms complex text into a program of operations ($P = \{o^1, \ldots, o^n\}$) using LLMs, which are then executed by various vision models (*e.g.*, object detectors, VQA models). Each operation $o^i$ yields an output $a_i$, where $a_i$ serves as the input for the next operation. The final prediction is made after all operations are executed. However, this approach highlights two key shortcomings of existing approaches: i) module error, the operation models can not predict the answer correctly; ii) planning error, the LLM might generate unfaithful reasoning.

**Framework overview** Fig. 1 depicts the overall framework. For each operation $o^i$, we get a set of candidate answers $\{a_1^i, \ldots, a_k^i\}$, with probabilities $\{p_1^i, \ldots, p_k^i\}$. Unlike VISPROG, which directly takes the top answer, we use additional verification modules to verify each candidate answer, thus producing verification scores $\{s_1^i, \ldots, s_k^i\}$. Then we take the verification score $s$ to calibrate the original scores. Additionally, we use the verification scores to search for a program with high verification scores, in order to refine the execution program $P$ by tree-searching.

## 3.2 Verification modules

The verification modules aims to verify the candidate answers $\{a_1^i, \ldots, a_k^i\}$ given an operation $o^i$. For example, the LOC(nightstand) operation returns a set of candidate bounding boxes containing a nightstand, then the verification module verifies whether each of the returned boxes contains a nightstand and produces verification scores. Our verification module is a mixture of three off-the-shelf sub-verifiers. The output scores of the three verifiers are combined as the final verification score. It is important to emphasize that the verification model does not incorporate any extra pre-trained models; instead, it utilizes the verifiers that are integral to the execution modules of VISPROG. To ensure equitable comparisons between modules, we have deliberately chosen these specific sub-verifiers.

**Image-text matching verifier** calculates the similarity between the whole images and all candidate sentences, which returns the semantic representation of the image-sentence pair. We construct the candidate sentences $\mathcal{T}_{ans}$ by filling the template "a photo of" with candidate answers. In this work, we select CLIP (Radford et al., 2021) to calculate the similarity between images and sentences, *i.e.*, $s_{ans}^{itm} = \text{ITM}(\mathcal{T}_{ans}, img)$.

**Image captioning verifier** leverages natural language to describe the visual details of the image. We first get the caption of the image $\mathcal{C}_{img}$ by BLIP (Li et al., 2022b). We then construct

the descriptions of candidate answers $\mathcal{C}_{ans}$ with the template "the image describe". Specifically, for candidate question-answer pairs, we initially transform the pair into a sentence before inserting it into the template. After that, we calculate the sentence semantic similarity (Reimers & Gurevych, 2019) between the captions and the constructed descriptions as the verification score, *i.e.*, $s_{ans}^{cap} = sim(\mathcal{C}_{ans}, \mathcal{C}_{img})$ .

**Visual question-answering (VQA) verifier** is more flexible than others, which offers us more opportunities to evaluate the advanced relationships between image and language, such as entailment and factual consistency. Slightly different from the other two types of models, for the VQA verifier, we design templates w.r.t. the neural modules. For example, we use "Is there any object in the image?" for the object detection model, and use "Does this part look like object ?" for the classification model used in the abstract reasoning task. We determine the verification score by BLIP (Li et al., 2022b) by calculating the difference in answer probabilities $\mathcal{Q}_{ans}$ between "yes" and "no"

$$s_{ans}^{vqa} = \text{VQA}(\mathcal{Q}_{ans}, True) - \text{VQA}(\mathcal{Q}_{ans}, False) \tag{1}$$

**Verification score**   After obtaining the scores from each individual verification module, the verification score is averaged over all scores for each given answer, *i.e.*, $s_{ans} = \text{avg}(s_{ans}^{itm}, s_{ans}^{cap}, s_{ans}^{vqa})$

**Negative sampling.** Empirically, we find that directly applying this verification score does not work well, because the score scales for different kinds of candidates are not well-calibrated. Motivated by recent works in truthfulness (Li et al., 2022a), commonsense (Ye et al., 2022), and bias (Ruggeri & Nozza, 2023), we propose to take the difference of a candidate answer $a_j$ with its semantic opposites $n_j$ as the final verification score. More specifically, the semantic opposite $n_j$ is selected based on the text embeddings from CLIP Radford et al. (2021), *i.e.* the word of lowest embedding similarity is selected. For example, the semantic opposite of "nightstand" is "stocking". We then compute the difference of the verification scores of the candidate answer and its semantic opposites, and get the final verification score. Mathematically, given a candidate answer $a^j$, the final verification score is $s_j = s_{a_j} - s_{n_j}$.

**Calibration using verification scores**   After obtaining the verification scores of all candidate answers $S = \{s_1, \ldots, s_k\}$, we normalize them as weights and calibrate the candidate predictions, $p'_j = w_j * p_j$, where $w_j$ is the normalized verification score. More specifically, the verification score $s_j$ is re-scaled to $w_j = \frac{s_j - s_{min}}{s_{max} - s_{min}} \cdot (\tau - \frac{1}{\tau}) + \frac{1}{\tau}$, where $\tau$ is a hyperparameter controlling the scaling factor $s_{min}, s_{max}$ are the minimum or maximum of all the candidate scores.

## 3.3   Exploration with reasoning trace

To mitigate the planning errors, we further apply the verification scores to refine the reasoning trace predicted by LLMs. Motivated by the recent works showing that searching through a compositional problem space can greatly improve the performance of LLMs for complex tasks (Yao et al., 2023; Khalifa et al., 2023; Hao et al., 2023), we introduce our dynamic reasoning trace searching procedure, which takes advantage of both the LLM self-correctness potential and our verification modules. In **Appendix A**, we show the complete algorithm of ExoVIP.

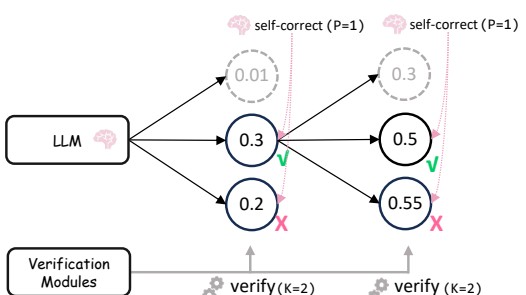

Figure 2: Search of the reasoning trace. We beam search through the program tree, based on the verification scores as well as the LLM self-correctness.

**Tree-based reasoning trace searching (TRS)**   The reasoning trace searching procedure is represented as a tree structure, where each node is a reasoning operation. To get a better

reasoning trace, we search from the tree using the beam search algorithm (Graves, 2012; Boulanger-Lewandowski et al., 2013; Sutskever et al., 2014), which has long been proven to be effective in sequence-to-sequence problems.

More specifically, our trace searching procedure contains two steps. First, in order to generate more diverse reasoning traces to search from, we randomly perturb the in-context examples in the prompt for LLM. Second, after we get the result of candidate neural modules, we sort them according to the verification scores and select the top $K$ candidate reasoning traces.

**Post-hoc self-correction (PSC)**   In some cases, the verification scores can be very close for the top-rated $K$ traces, which could result in suboptimal. Inspired by the zero-shot ranking ability of LLM Hou et al. (2023), we further use the self-correctness ability of LLMs to rank the $K$ traces and select the top $P$ from them ($P < K$). More details of the prompts used for LLM self-correction are included in Sec. E.2.

## 4   Experiments

In this section, we apply the ExoViP to VisProg, demonstrating the effectiveness of our approach through results and anlysis derived from six distinct tasks, including visual question answering, referring expression understanding, visual reasoning using natural language, abstract reasoning, language-guided image editing, video question answering. Subsequent to this, we delve into an exploration of potential future projections in Sec. 4.5. For additional information regarding the implementation and experiment setups and selection of baselines, please refer to **Appendix E**.

### 4.1   Compositional Visual Question Answering

#### 4.1.1   Main Results and Analysis

We evaluate the efficacy of ExoViP on a compositional visual question answering task GQA (Hudson & Manning, 2019), and benchmark against top vision-language models, including BLIP2-flant5-xxl (Li et al., 2023b), InstructBLIP-flant5-xl (Dai et al., 2023), and use LLaVA-1.5-13B (Liu et al., 2023a) as a reference point. Our method boosts VisProg's score from 57.41 to 61.49, surpassing BLIP2 and InstructBLIP, as detailed in Tab. 1. It's important to note that our method does not incorporate any additional modules or knowledge compared to VisProg. The verification modules

| | Methods | Accuracy |
|---|---|---|
| | BLIP2-xxl (Li et al., 2023b) | 49.20 |
| | InstructBLIP-flant5-xl (Dai et al., 2023) | 55.39 |
| | Llava-1.5-13b* (Liu et al., 2023a) | 74.56 |
| 0 | VisProg (Gupta & Kembhavi, 2023) | 57.41 |
| 1 | ExoViP w/o self-correctness & negative sampling & search | 57.11 |
| 2 | ExoViP w/o self-correctness & search | 58.53 |
| 3 | ExoViP w/o self-correctness (TRS) | 60.57 |
| 4 | ExoViP w/o verification (PSC) | 60.16 |
| 5 | ExoViP | 61.49 |

Table 1: Results of compositional visual question answering on GQA. Llava-1.5-13b* is tuned on GQA training corpora, and evaluated with additional prompt.

that we use are inherent to VisProg itself. To verify the effectiveness of each component in our method, we run a series of ablation studies on our framework (also in Tab. 1). We have the following observations:

**Negative sampling enhances the robustness of ExoViP**   Merely adding verification modules to a system (Line-1) is not sufficient for achieving better results; it may actually lead to decreased performance. On the other hand, when we implement a negative sampling technique that utilizes semantic opposites in these verification modules (Line-2), there is a marked enhancement in the system's performance. We posit that this approach could be instrumental in reducing the likelihood of new errors being introduced.

**TRS effectively utilizes the verification score**   Empirical evidence shows that implementing TRS can increase accuracy from an initial 58.53 to a subsequent 60.57 (Line-3). This improvement in precision underscores the effectiveness of our verification-based search strategy, which has the potential to resolve numerous planning errors.

**Verification mechanism enhance LLM self-correction**   In Line-4, we exclusively utilize the post-hoc self-correction (PSC) of the LLM during the process of trace searching, eschewing the use of verification scores. The findings demonstrate an enhancement in accuracy of 2.75 when compared to the original VISPROG. However, the implementation of both verification scores and LLM self-correctness concurrently results in a superior performance enhancement.

### 4.1.2   Analysis Experiments

We experiment to explore how the tree-based search algorithm and verification scores interact to improve our approach. The algorithm uses scores to guide branching, while insights from the search enhance score contrasts, refining path differentiation and aiding in finding the best solution. All experiment settings are aligned with the Appendix B.

**Mixture of Sub-verifers.**   We evaluate the effects of different types of verification modules with the setting of the best demonstration setting. As is illustrated in Tab. 2, Different verification modules share similar boost gain, but a mixture of these modules can benefit more.

**Enhanced Verification through TRS.**   Our analysis, illustrated in Fig. 6, shows that implementing our trace-searching strategy significantly improved verification scores. Additionally, the increased variance in scores suggests our method could further refine the effectiveness of reasoning traces.

| Methods | Accuracy |
|---|---|
| Base | 58.14 |
| Image-text Matching | 59.26 |
| Image Caption | 59.22 |
| Visual QA | 59.35 |
| All | 60.03 |

Table 2: Analysis on the sub-verifiers.

**Enhanced Planning Efficacy**   In our comparative analysis of the GQA task, we observed that our method, EXOVIP, significantly outperforms the baseline, VISPROG, in terms of planning efficiency and accuracy. The average number of planning steps required by EXOVIP decreased from 5.92 to 4.77, indicating that the TRS strategy employed by our method streamlines the planning process, allowing for a more direct path to the final plan. We also compute the average inference time, which is shown on **Appendix D**. Furthermore, we noted a reduction in the error rate, with the percentage of unexecutable plans dropping from 5.84% to 3.82%. This demonstrates that EXOVIP not only reduces the complexity of the planning process but also enhances the reliability of the generated plans, predicting a higher number of executable routines compared to the baseline.

### 4.2   Abstract Visual Reasoning

We tested model performance on the KILOGRAM (Ji et al., 2022) dataset's text-to-image retrieval task, involving 1,251 tangram puzzles with abstract shape recognition. Accuracy was the main evaluation metric. The CLIP-large model (Radford et al., 2021) served as our baseline for the text-to-image retrieval task. As is illustrated in Fig. 18, in our approach, we leverage the LLM to identify potential semantic components of a given description. Simultaneously, we segment the image into distinct visual parts. Following this, we align the identified semantic parts with their corresponding visual counterparts to optimize the matching process. Tab. 4 illustrates how our method effectively utilizes conceptual components to enhance abstract image understanding. However, a performance gap is still noticeable when compared to CLIP. Despite our method narrowing this gap, it is still unable to reach SOTA performance levels. The importance of part identification in human abstraction has been well-established in prior research (Tversky & Hemenway, 1984). We posit that the efficacy of our approach could be significantly improved by integrating a more advanced scene segmentation model.

Table 3: Visual referring expression on Ref-COCO, RefCOCO+, and RefCOCOg.

| Methods | IoU |
|---|---|
| Qwen-vl-chat-7b (Bai et al., 2023) | 32.54 |
| VISPROG (Gupta & Kembhavi, 2023) | 27.28 |
| EXOVIP | 31.50 |

Table 4: Abstract reasoning on KILO-GRAM.

| Methods | Accuracy |
|---|---|
| CLIP-large (Radford et al., 2021) | 27.26 |
| VISPROG (Gupta & Kembhavi, 2023) | 24.46 |
| EXOVIP | 26.22 |

Table 5: Visual reasoning on NLVR2.

| Methods | Accuracy |
|---|---|
| OFA-large (Wang et al., 2022) | 58.38 |
| VISPROG (Gupta & Kembhavi, 2023) | 67.66 |
| EXOVIP | 67.96 |

Table 6: Image editing on MagicBrush.

| Methods | CLIP-I | DINO |
|---|---|---|
| InstructPix2Pix (Brooks et al., 2022) | 84.19 | 69.60 |
| VISPROG (Gupta & Kembhavi, 2023) | 90.82 | 82.70 |
| EXOVIP | 91.27 | 83.40 |

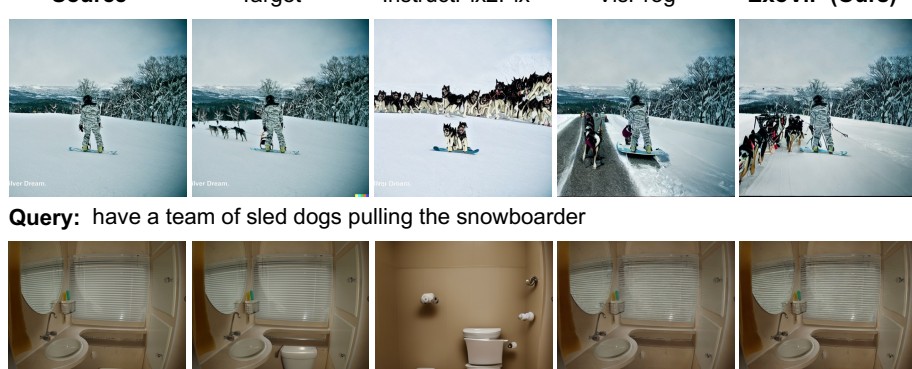

**Query:** have a team of sled dogs pulling the snowboarder

**Query:** open the lid of a toilet

Figure 3: Qualitative results of text-guided image editing on MagicBrush

## 4.3 Language-grounded Visual Tasks

**Visual Referring Expressions (VRE)**   Our study on VRE used a subset of the RefCOCO, RefCOCO+, and RefCOCOg datasets (Yu et al., 2016; Kazemzadeh et al., 2014), and evaluated using intersection-over-union (IoU). We benchmarked against the Qwen-vl-chat-7b (Bai et al., 2023) model, a high-performing, pre-trained vision-language model. The results presented in Tab. 3 illustrate that, even though our approach does not reach the state-of-the-art (SOTA) performance achieved by Qwen-vl on the RefCOCO dataset, it nonetheless narrows the gap between VISPROG and large vision-language models. Qwen-vl is a highly complex model, constructed on a language learning model (LLM) consisting of 7 billion parameters and trained on a corpus of trillions of tokens. In contrast, our approach utilizes a team of specialized experts, whose combined parameters amount to less than 1 billion. We are optimistic that the performance of our method can be further enhanced by incorporating more sophisticated experts.

**Natural Language Visual Reasoning**   In this work, we use the NLVR2 (Suhr et al., 2019) balanced test set for evaluation and use the OFA-large (Wang et al., 2022) as our baseline, unlike many multi-modal language models that struggle with dual-image inputs. Tab. 5 presents our findings. While VISPROG demonstrates a strong capability for complex reasoning compared to the end-to-end model, our method struggles to enhance its performance significantly. We attribute this to our sole reliance on VQA modules for solving NLVR problems. Specifically, the efficacy of the decomposed VQA steps is intrinsically constrained by the performance of the VQA model itself. This limitation becomes especially troublesome when errors accumulate over a series of VQA steps, consequently hamper-

ing the overall performance. As a path forward, we foresee potential advancements in the planning process, which could involve the integration of a wider array of expert inputs.

**Text-guided Image Editing** We use the MagicBrush dataset (Zhang et al., 2023a)for evaluation. Image quality is gauged using CLIP-I and DINO embeddings for similarity assessment. Our baseline is the GPT3-augmented InstructPix2Pix (Brooks et al., 2022) model. The results from both CLIP-I and DINO are presented in Tab. 6. These results illustrate the capability of our method to enhance the similarity between the edited image and the target image, signifying the precision of our image editing technique. For a more comprehensive evaluation of the editing quality, we have conducted a case study. Fig. 3 exhibits some instances using MagicBrush. It is observed that non-compositional methods, *i.e.* Instruct-Pix2Pix, tend to alter unrelated pixels, whereas compositional methods, *i.e.* VISPROG and our model, offer more control. Furthermore, when compared to VISPROG, our method excels in two key areas: accurately pinpointing the region that requires editing, and adjusting the image to the appropriate extent. This demonstrates the superiority of our method in both localization and modification of the image.

## 4.4 Spatial-Temporal Video Reasoning

We conducted experiments using the subset of AGQA 2.0 dataset (Grunde-McLaughlin et al., 2022). Our reference was the Video-LLaVA (Lin et al., 2023), a top-tier vision-language model known for its superior performance on numerous benchmarks. In our methodology, we address the question by breaking it down into temporal and spatial components. For the temporal aspect, we aim to find the event or ac-

| Methods | Accuracy |
|---|---|
| Video-LLaVA (Lin et al., 2023) | 30.38 |
| EXOVIP w/o verification | 37.88 |
| EXOVIP | 38.00 |

Table 7: Results of Spatial-Temporal Reasoning on AGQA.

tion within a video. This is achieved by uniformly sampling frames from the video and generating corresponding captions. We then compute the sentence similarity between these captions and the input query. Subsequently, we identify the event by locating the video segment with the highest similarity, utilizing a monotonic stack algorithm. By adopting this approach, we can effectively mitigate the Out-of-Vocabulary (OOV) issue that plagues current action classification models. Regarding the spatial component, it is predominantly addressed by existing VQA models. The experimental outcomes, as presented in Tab. 7, indicate that the compositional method yields strong performance. However, the benefits brought by verification are limited. Upon further examination of the results and the underlying reasoning paths, we observe that the majority of the unsuccessful cases can be attributed to the performance of the VQA models, a trend that aligns with findings from the NLVR task.

## 4.5 Discussion

### 4.5.1 Error analysis of VISPROG and EXOVIP

We manually analyze 100 randomly sampled failure cases on VISPROG. We find that there are three typical reasons for the failures: (a) vision module prediction error; (b) LLM planning error; (c) others. We demonstrate the statistics of the failure cases in Fig. 4 (left). Following the application of our proposed framework, we reassessed the same cases in **??** (right) and were pleased to discover a reduction in module errors by 28.87%, and a decrease in planning errors by 42.35%. Nevertheless, our current strategy was unable to rectify 69.8% of the errors. When juxtaposed with the data from Tab. 1, our method has enhanced VISPROG by 7.11%, which is lower than the improvement of the failure cases. This outcome suggests that our approach may give rise to novel challenges. We further demonstrate common errors of our method in Fig. 9 and Fig. 10. We find the majority of these failure cases originate from the VQA module.

**Additional error analysis** Our methodology acknowledges the possibility that the inclusion of verifiers might inadvertently increase the error rate. To counteract this, we have

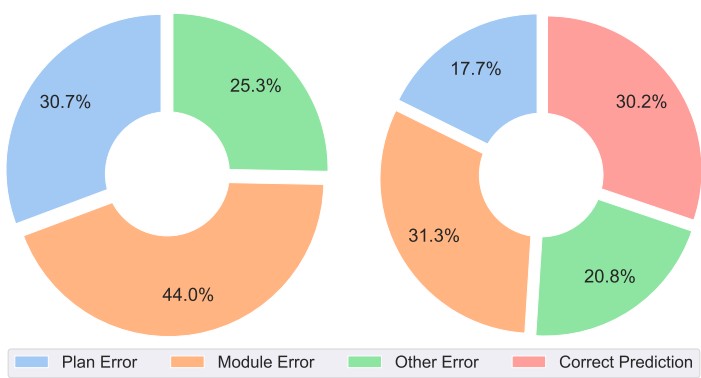

Figure 4: Distribution of the failure cases of original VISPROG (left), and distribution of the failure cases of EXOVIP (right)

adopted a negative sampling approach, and we have integrated with a verification score and the inherent self-corrective feature of LLMs. The efficacy of these combined strategies in reducing the incidence of additional errors is evidenced by the results displayed in Tab. 2. Nonetheless, while our approach successfully diminishes errors related to planning and module execution, it can occasionally lead to the introduction of new errors. Moving forward, we aim to enhance our system by incorporating a greater number of verifiers to more effectively resolve these issues.

### 4.5.2 Method generalizability

To validate the generalizability of our method, we applied it to ViperGPT, which composes available modules by generating Python codes. We equip ViperGPT with our method and test its performance on the GQA dataset. The results, presented in Tab. 8, reveal a less significant performance boost compared to VISPROG. We attribute this to ViperGPT providing only a few demonstration examples and adjusting the parameters of the code-generation model to deterministically generate subroutines. We believe this could be improved by introducing diverse demonstrations, similar to VISPROG.

| Methods | Accuracy |
|---|---|
| ViperGPT (Dídac et al., 2023) | 45.47 |
| ViperGPT+ExoViP | 46.84 |

Table 8: Results for ViperGPT on GQA.

## 5 Conclusion

In this work, we identify two key types of errors in existing compositional methods: planning errors and module errors. To address these errors, we introduce an innovative verification framework EXOVIP. This framework verifies the correctness of vision module predictions. It corrects module errors by calibration and refines the planning process through tree searching. During this process, it considers both verification scores and the self-correctness of LLM. Applying the EXOVIP to two existing models, we achieve performance improvements across five different tasks. The results reinforce the promise and potential of EXOVIP on various open-world compositional reasoning tasks, marking an important milestone in the realm of multi-modal tasks involving complex reasoning.

**Acknowledgments** The authors thank the reviewers for their insightful suggestions to improve the manuscript. This work presented herein is supported by the National Natural Science Foundation of China (62376031).

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

# Appendices

## Table of Contents

## A  Exoskeleton Algorithm

We demonstrate the overall algorithm of our method in Algorithm 1. There are mainly two parts: step-by-step verification and exploration with reasoning trace. To be more specific, we fuse the self-correctness ability of LLM into the procedure of tree-based reasoning trace searching, which has shown potential in calibrating the effectiveness of the searching algorithm.

## B  Proof-of-concept Pilot Experiments

To evaluate the effectiveness of the verification modules, we try to find the relationship between verification scores and accuracy. All experiments are applied to the GQA dataset. We first disturb the examples in the demonstrations to get different plan results and corresponding verification scores. Specifically, we change the order of examples and select different portions of examples with four settings. After evaluation, we calculate the mean of verification scores of all steps. As is shown in Fig. 5, we are delighted to find the verification scores positively contribute to final accuracy. However, the trend is decreasing, which means when the verification scores increase to a certain extent, higher verification scores do little contribution to the final accuracy.

---

**Algorithm 1:** Exoskeleton Algorithm

---

**Input:** start step ($e_0$), goal node ($g$), scaling factor ($\tau$), verification size ($K$), rank size ($P$)
**Output:** Verified reasoning trace and intermedia results
$openList \leftarrow e_0$
$closedList \leftarrow empty\ list$
$path \leftarrow empty\ list$
**while** *open list is not empty* **do**
    $sort(openList, key = e_s)$
    Select top $K$ steps from $openList$ and put it in $closedList$ and empty $openList$
    $rank(closedList, key = LLM(e))$
    Select top $P$ steps to update $closedList$
    **for** *e in closedList* **do**
        **if** *e is g* **then**
            $path.add(e)$
            return *path*
        **else**
            $openList.add(e.next)$
        **end**
    **end**
    **for** *e in openList* **do**
        $e_s = avg(e_s^{item} - e_n^{item}, e_s^{cap} - e_n^{cap}, e_s^{vqa} - e_n^{vqa})$
        $e \leftarrow Verify(NORM(e_s, \tau), e)$
    **end**
**end**

---

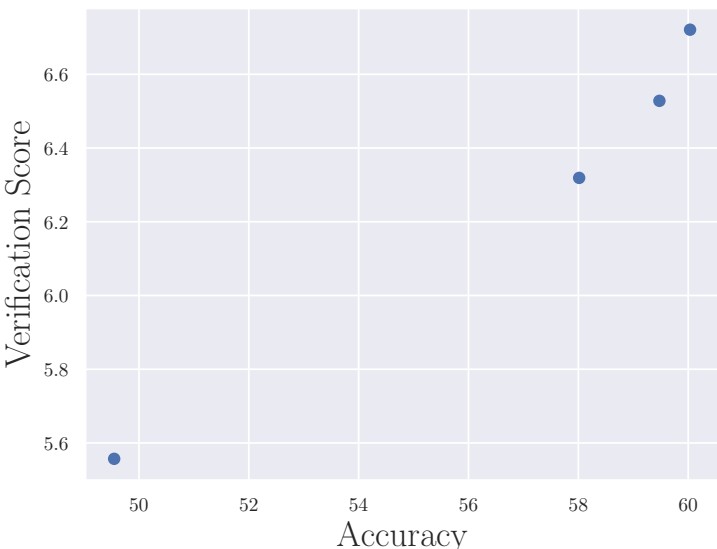

Figure 5: Accuracy on GQA positively correlates with the verification scores.

## C  Enhanced Verification through TRS

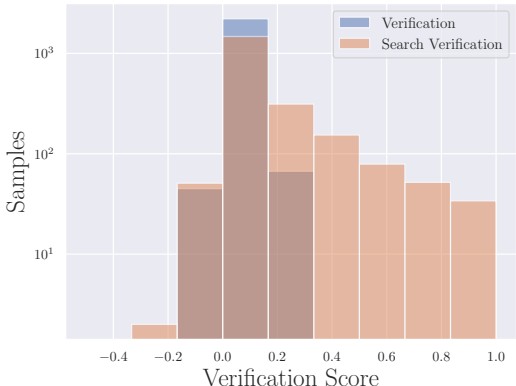

Figure 6: Distribution of verification scores w. and w/o trace searching.

In 6, we present a graphical representation illustrating the relationship between the trace-searching strategy and the verification score. The x-axis quantifies the verification score associated with each trace, while the y-axis denotes the number of traces corresponding to each score.

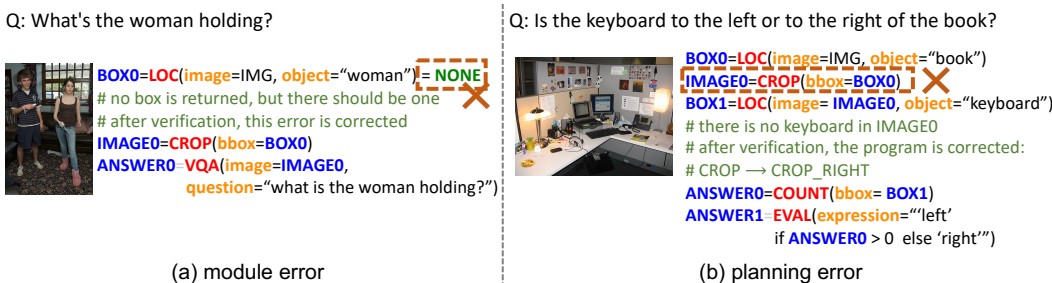

Figure 7: Existing methods suffer from two types of errors: (a) vision module prediction error and (b) LLM planning error. Our verification modules help correct the errors.

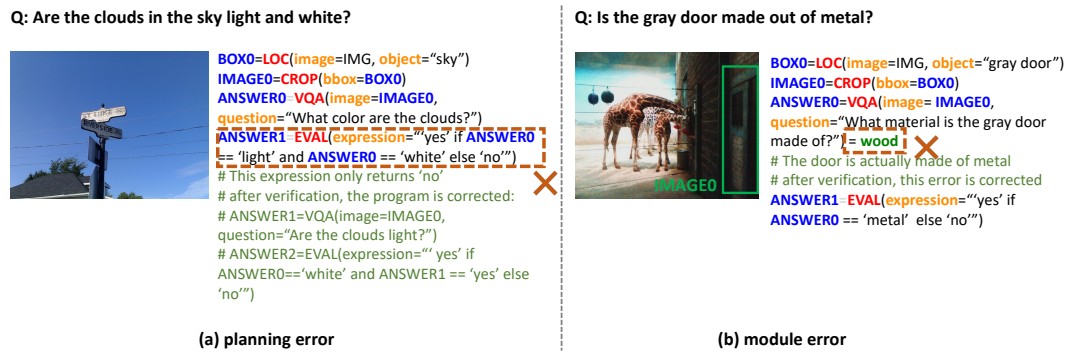

Figure 8: More examples of the two types of errors: (a) vision module prediction error and (b) LLM planning error.

In Figs. 7 and 8, we show examples of failure cases of the original VISPROG.

**Q: Are both the shoe and the cloud the same color?**

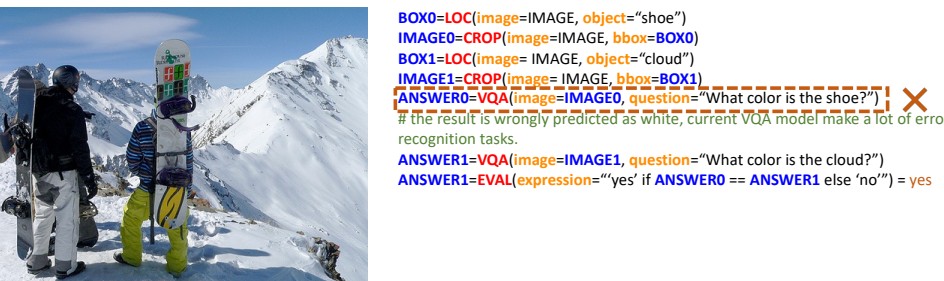

**BOX0**=**LOC**(**image**=IMAGE, **object**="shoe")
**IMAGE0**=**CROP**(**image**=IMAGE, **bbox**=**BOX0**)
**BOX1**=**LOC**(**image**= IMAGE, **object**="cloud")
**IMAGE1**=**CROP**(**image**= IMAGE, **bbox**=**BOX1**)
**ANSWER0**=**VQA**(**image**=**IMAGE0**, **question**="What color is the shoe?")  ✗
# the result is wrongly predicted as white, current VQA model make a lot of errors on color recognition tasks.
**ANSWER1**=**VQA**(**image**=**IMAGE1**, **question**="What color is the cloud?")
**ANSWER1**=**EVAL**(**expression**="'yes' if **ANSWER0** == **ANSWER1** else 'no'") = yes

Figure 9: Common failure cases: some modules perform badly on certain tasks, *e.g.* the VQA module performs poorly on color recognition tasks.

**Q: Which place is it ?**

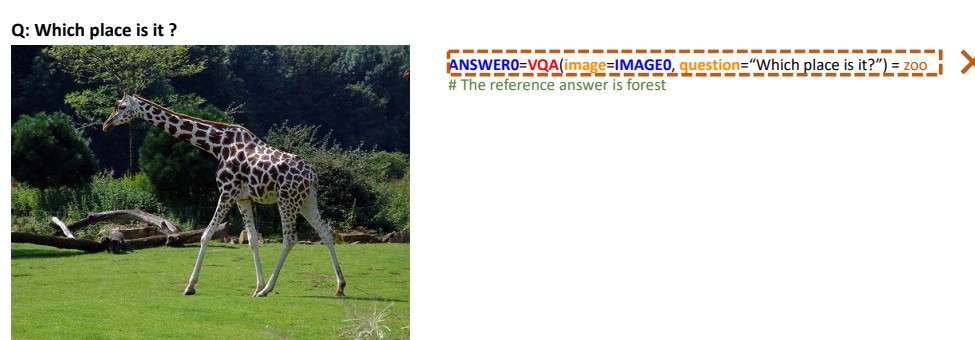

**ANSWER0**=**VQA**(**image**=**IMAGE0**, **question**="Which place is it?") = zoo  ✗
# The reference answer is forest

Figure 10: Common failure cases: some queries can not be decomposed into sub-tasks. Our method helps little with these non-decomposable queries.

## D   Efficiency Analysis

We present the average inference time on the GQA dataset. Generally, the temporal expenditure of our tree-based search method significantly surpasses that of VisProg. However, the majority of the time is consumed by the call of the OPENAI API, an issue we posit is intrinsic to analogous works Yao et al. (2023); Feng et al. (2023); Zhou et al. (2023). When compared to Depth First Search/ Breadth First Search Yao et al. (2023) or Monte Carlo Tree Search Feng et al. (2023); Zhou et al. (2023), we assert that our beam search-based method can achieve an optimal equilibrium between efficiency and effectiveness. In addition, when comparing with End-to-End model, we find that the most significant contributor to the overall time cost is the Planning Time. We have determined that this delay is largely attributable to Internet latency, as our system utilizes the GPT-3.5-turbo API. We are confident that this latency can be mitigated by deploying the LLM locally, which would reduce the dependency on network response times. Additionally, we are exploring ways to enhance the parallelism of our system's submodules, which we believe will further improve efficiency.

Table 9: Average Inference Time on the GQA Dataset

| Methods | Total Infer. Time (s) | Planning Time (s) | Module Infer. Time (s) |
|---|---|---|---|
| BLIP2-Flant5-xxl | 0.17 | - | 0.17 |
| LLaVA-1.5-7B | 0.45 | - | 0.45 |
| VISPROG | 1.59 | 1.10 | 0.49 |
| EXOVIP | 4.32 | 3.64 | 0.68 |

# E  Implementation details

## E.1  Visual modules.

| Task | Operation Modules | | | | | Verification Modules | |
|---|---|---|---|---|---|---|---|
| Compositional Image QA | **LOC**
OWL-ViT | **VQA**
BLIP | **FILTER**
CLIP | **COUNT**
len() | **EVAL**
eval() | **SIM**
CLIP | **CAP**
BLIP |
| | **CROP**
PIL.crop() | **CROPLEFT**
PIL.crop() | **CROPRIGHT**
PIL.crop() | **CROPABOVE**
PIL.crop() | **CROPBELOW**
PIL.crop() | **VQA**
BLIP | |
| Visual Grounding | **LOC**
OWL-ViT | **FILTER**
CLIP | **TAG**
PIL.rectangle() | | | **SIM**
CLIP | |
| Natural Language for Visual Reasoning | **VQA**
BLIP | **EVAL**
eval() | | | | **CAP**
BLIP | **VQA**
BLIP |
| Abstract Reasoning | **PART**
ChatGPT | **SEG**
Maskformer | **ALIGN**
CLIP | **SELECT**
CLIP | | **SIM**
CLIP | |
| Text-guided Image Editing | **SEG**
Maskformer | **SELECT**
CLIP | **REPLACE**
Stable Diffusion | | | **SIM**
CLIP | |

Figure 11: The neural modules (green) and symbolic modules (pink) used in our experiments.

We summarize the operation modules and the verification modules of different tasks in Fig. 11. In practice, the candidate neural modules include OWL-ViT (Minderer et al., 2022), CLIP (Radford et al., 2021), BLIP (Li et al., 2022b), ChatGPT, MaskFormer (Cheng et al., 2021), Stable Diffusion (Rombach et al., 2022). In order to validate the effectiveness of our method and eliminate the benefits of external knowledge such as more advanced vision-language models which are trained on larger datasets. Both operation modules and verification modules are selected from the same candidate neural module sets. In other words, not all modules are verified on the mixture of all three types of modules.

## E.2  LLM Prompts

We demonstrate the prompt for self-correctness of all five tasks.

```
You are a ranker for a planner who use the candidate modules to answer a question:
QUESTION, select the best solutions for answering the question
candidate modules include: LOC: detection, VQA: visual question answering, EVAL: use
logic operation, RESULT: wrap up the final result,
CROP/CROP_LEFTOF/CROP_RIGHTOF/CROP_FRONTOF/CROP_INFRONT/CROP_INFRONTOF/CROP_BEHIND/CROP_
AHEAD/CROP_BELOW/CROP_ABOVE: crop the image.
Current solutions:
0 PLAN
1 PLAN
If the modules in the solutions have better cause-and-effect relations, and more likely
to answer the question, please rank it first. If you are unsure, please keep the
original rank. Return sequence number of currently best solution, for example 0,1,2,3,
DO NOT RETURN ANYTHING ELSE EXCEPT FOR NUMBERS SPLIT by ,
```

Figure 12: Self-correctness prompt of compositional question answering.

```
You are a ranker for a planner who use the candidate modules to carry out the
instruction: QUESTION, select the best solutions for carrying out the instruction
candidate modules include: LOC: detection, FILTER: filter unrelated objects, TAG: tag
the object, RESULT: wrap up the final result
Current solutions:
0 PLAN
1 PLAN
If the modules in the solutions have better cause-and-effect relations, and more likely
to answer the question, please rank it first. If you are unsure, please keep the
original rank. Return sequence number of currently best solution, for example 0,1,2,3,
DO NOT RETURN ANYTHING ELSE EXCEPT FOR NUMBERS SPLIT by ,
```

Figure 13: Self-correctness prompt of visual grounding.

```
You are a ranker for a planner who use the candidate modules to evaluate the statement:
QUESTION, select the best solutions for evaluating the statement
candidate modules include: VQA: visual question answering, EVAL: use logic operation,
RESULT: wrap up the final result \n Current solutions
Current solutions:
0 PLAN
1 PLAN
If the modules in the solutions have better cause-and-effect relations, and more likely
to answer the question, please rank it first. If you are unsure, please keep the
original rank. Return sequence number of currently best solution, for example 0,1,2,3,
DO NOT RETURN ANYTHING ELSE EXCEPT FOR NUMBERS SPLIT by ,
```

Figure 14: Self-correctness prompt of natural language for visual reasoning.

```
You are a ranker for a planner who use the candidate modules to carry out the
instruction: QUESTION, select the best solutions for carrying out the instruction
candidate modules include: SEG: segmentation, SELECT: select most related object,
REPALCE: edit image, RESULT: wrap up the final result
Current solutions:
0 PLAN
1 PLAN
If the modules in the solutions have better cause-and-effect relations, and more likely
to answer the question, please rank it first. If you are unsure, please keep the
original rank. Return sequence number of currently best solution, for example 0,1,2,3,
DO NOT RETURN ANYTHING ELSE EXCEPT FOR NUMBERS SPLIT by ,
```

Figure 15: Self-correctness prompt of text-guided image editing.

```
You are a ranker for a planner who use the candidate modules to carry out the
instruction: QUESTION, select the best solutions for carrying out the instruction
candidate modules include: PART: take apart an object, SEG: segment, ALIGN: align object
with query, RESULT: wrap up the final result
Current solutions:
0 PLAN
1 PLAN
If the modules in the solutions have better cause-and-effect relations, and more likely
to answer the question, please rank it first. If you are unsure, please keep the
original rank. Return sequence number of currently best solution, for example 0,1,2,3,
DO NOT RETURN ANYTHING ELSE EXCEPT FOR NUMBERS SPLIT by ,
```

Figure 16: Self-correctness prompt of visual abstract reasoning.

## E.3    Details of Visual Abstract Reasoning

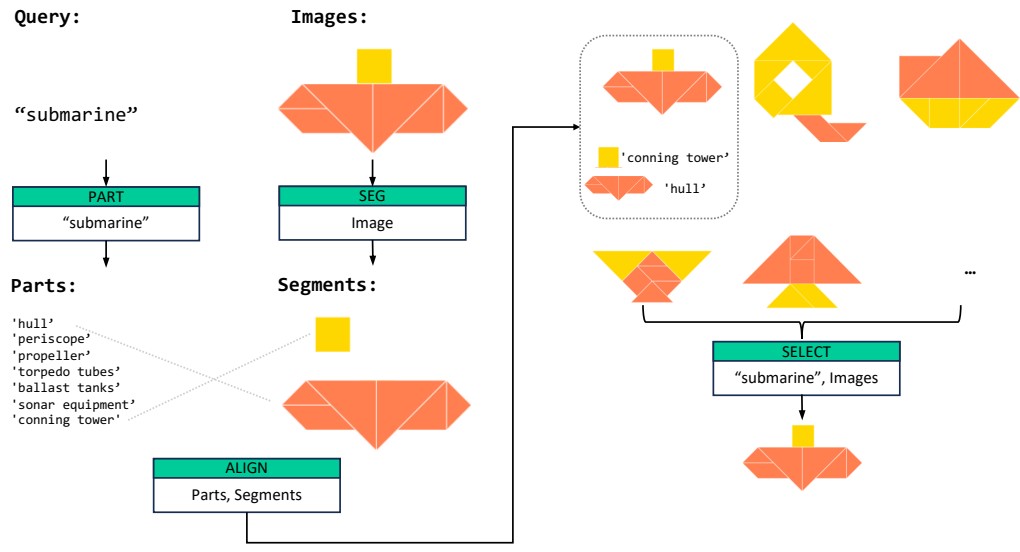

Figure 17: Implementation of abstract reasoning.

In Fig. 17, we demonstrate our implementation of compositional methods on KILOGRAM dataset. Given an image, we segment it into several parts. At the same time, we adopt LLM to parse the query to several components. After that, we match the visual and textual components by their semantic similarity. Finally, we take the alignment score to retrieve the best matched image.

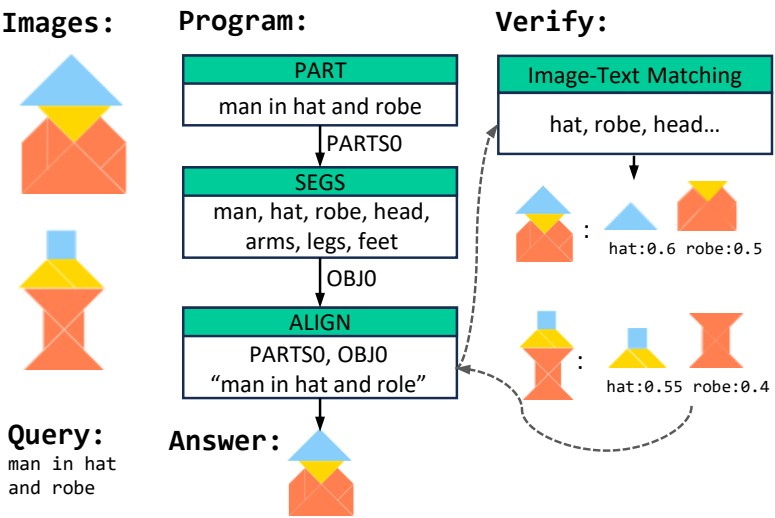

Figure 18: Implementation of EXOVIP on abstract reasoning.

### E.4 Implementation details

In practice, for the verification modules, we set the $\tau$ as 2.0 for $LOC$ module, 1.5 for $SELECT$ module, and $ALIGN$ module, 1.2 for other modules. For the negative sampling strategy, we select words sharing semantic similarity less than 0.5 to construct the semantic opposite vocabulary and randomly sample one semantic opposite for each answer. In the searching process, we set up $K$ as 4, and $P$ as 2. To improve the efficiency of our search algorithm, we set the branching factor as 3. To make the comparison fair, we use the same or fewer examples in the prompts for our methods, and select the verification modules from the operation modules. We apply our experiments on NVIDIA A100 GPU and NVIDIA 3090Ti GPU.

## F Qualitative study.

### F.1 Qualitative examples

We additionally exhibit more examples that can be improved by our method. As is shown in these examples, all five types of tasks could be further improved by our framework.

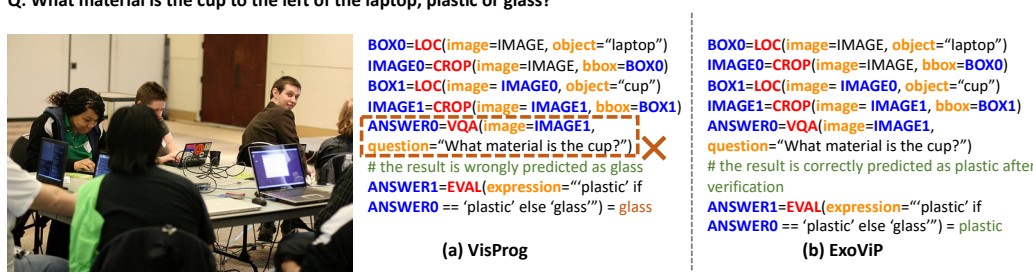

Figure 19: Qualitative study for GQA.

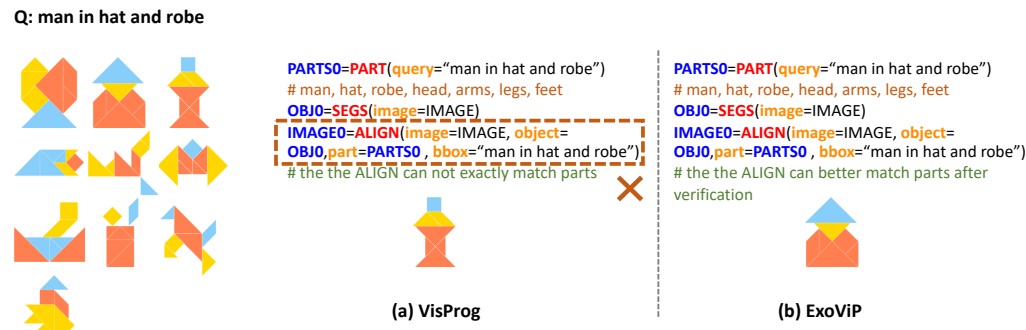

Figure 20: Qualitative study for KILOGRAM.

**Q: Tag the left zebra**

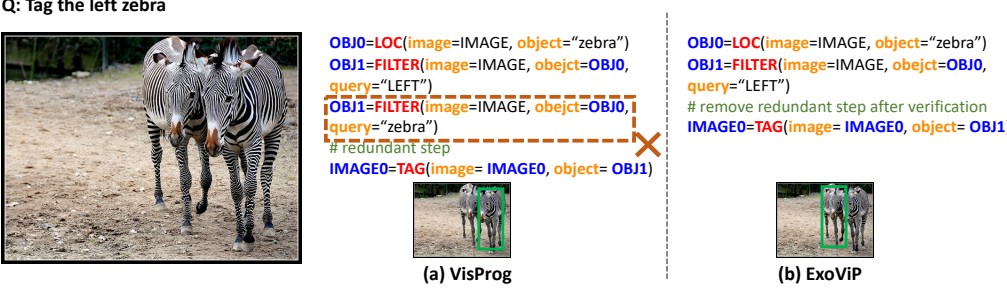

**OBJ0**=**LOC**(**image**=IMAGE, **object**="zebra")
**OBJ1**=**FILTER**(**image**=IMAGE, **obejct**=**OBJ0**, **query**="LEFT")
**OBJ1**=**FILTER**(**image**=IMAGE, **obejct**=**OBJ0**, **query**="zebra")
# redundant step
**IMAGE0**=**TAG**(**image**= IMAGE0, **object**= OBJ1)

**(a) VisProg**

**OBJ0**=**LOC**(**image**=IMAGE, **object**="zebra")
**OBJ1**=**FILTER**(**image**=IMAGE, **obejct**=**OBJ0**, **query**="LEFT")
# remove redundant step after verification
**IMAGE0**=**TAG**(**image**= IMAGE0, **object**= OBJ1)

**(b) ExoViP**

Figure 21: Qualitative study for RefCOCO.

**Q: The left image features a single fur-trimmed fingerless mitten with small embellishments dotting its front, and the right image shows a pair of fur-trimmed half-mitts with no thumb part showing.**

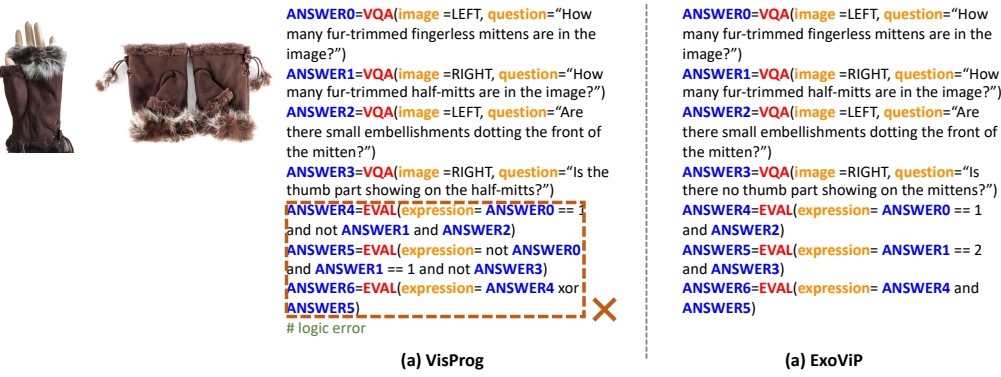

**ANSWER0**=**VQA**(**image** =LEFT, **question**="How many fur-trimmed fingerless mittens are in the image?")
**ANSWER1**=**VQA**(**image** =RIGHT, **question**="How many fur-trimmed half-mitts are in the image?")
**ANSWER2**=**VQA**(**image** =LEFT, **question**="Are there small embellishments dotting the front of the mitten?")
**ANSWER3**=**VQA**(**image** =RIGHT, **question**="Is the thumb part showing on the half-mitts?")
**ANSWER4**=**EVAL**(**expression**= ANSWER0 == 1 and not **ANSWER1** and **ANSWER2**)
**ANSWER5**=**EVAL**(**expression**= not **ANSWER0** and **ANSWER1** == 1 and not **ANSWER3**)
**ANSWER6**=**EVAL**(**expression**= ANSWER4 xor **ANSWER5**)
# logic error

**(a) VisProg**

**ANSWER0**=**VQA**(**image** =LEFT, **question**="How many fur-trimmed fingerless mittens are in the image?")
**ANSWER1**=**VQA**(**image** =RIGHT, **question**="How many fur-trimmed half-mitts are in the image?")
**ANSWER2**=**VQA**(**image** =LEFT, **question**="Are there small embellishments dotting the front of the mitten?")
**ANSWER3**=**VQA**(**image** =RIGHT, **question**="Is there no thumb part showing on the mittens?")
**ANSWER4**=**EVAL**(**expression**= ANSWER0 == 1 and **ANSWER2**)
**ANSWER5**=**EVAL**(**expression**= ANSWER1 == 2 and **ANSWER3**)
**ANSWER6**=**EVAL**(**expression**= ANSWER4 and **ANSWER5**)

**(a) ExoViP**

Figure 22: Qualitative study for NLVR2.

