# OpenReview forum: "ExoViP: Step-by-step Verification and Exploration with Exoskeleton Modules for Compositional Visual Reasoning"
_colmweb.org/COLM/2024/Conference — COLM_

### Official Review · Reviewer_MW2q · 2024-05-09

**Rating:** 5
**Confidence:** 4
**Ethics Flag:** 1

**Summary:**

The paper proposes two plug-and-play components to alleviate limitations in existing vision programming approaches with LLMs for compositional visual reasoning.
First, to improve accuracy of visual execution modules, it reuses visual modules in visual programs as additional verification modules. Negative sampling is proposed to calibrate scores of the verification modules.
Second, to reduce LLM planning mistakes, it uses tree-based reasoning trace searching and selects the best path.
Experimental results on five tasks demonstrate the effectiveness of the proposed method over the baseline VisProg.

**Questions To Authors:**

See reasons to reject.

**Reasons To Accept:**

- The proposed two components to alleviate mistakes in visual module execution and LLM planning are intuitive and interesting.
- The effectiveness of the proposed approach is demonstrated on five benchmark datasets and two baseline methods.

**Reasons To Reject:**

1. The negative sampling for score calibration is a bit strange. It uses an “antonym” for each answer for calibration. However, the way of selecting antonyms is simply using cosine similarity of text embeddings. Why does it make sense? It seems strange that the antonym of “night- stand” is “stocking”.

2. The algorithm to reduce planning mistakes is not clear. For example, how does the model sample K steps? How do the verification modules measure the plans?

3. Some ablations are missing.
- The hyper-parameters are not evaluated, such as the scaling factor \tau of the verification scores, the number of selected steps K and P.
- The proposed algorithm 1 should compare with simple approaches such as aggregating multiple paths from LLMs with different initial seeds etc.

4. The presentations of table 1 and 2 are confusing.
- The names of different variants of the EXOVIP model are not straightforward to know what the variant is. The textual descriptions sec 5.1.1 help understand these variants but it is still less clear for Line 3 and 4. Line 3 in Table 3 is a model without TRS but the description in Page 7 says it is a model simply with TRS. Similar for Line 4.
- The numbers in table 2 do not match any numbers in Table 1. Why?

---

> ### Author Rebuttal · Authors · 2024-05-31
>
> Thank you for your constructive comments
>
> > W1: strange negative sampling.
>
> **A:** The score computation, influenced by recent research in truthfulness, commonsense, and bias (Section 4.1), is a verifier-derived confidence score that becomes a normalized score through the inclusion of negative samples, enhancing system robustness and accuracy, and allowing unbiased model-blind confidence assessment of planning traces. The term "antonym" is defined by the greatest distance in VL semantic space using the CLIP text encoder, leading to unconventional pairings like "night-stand" and "stocking" based on computational results rather than human knowledge.
>
>
> > W2: The algorithm to reduce planning mistakes is not clear. (k step, measure plan)
>
>
> **A:** We manually assessed the results of applying our algorithm to VisProg, confirming its effectiveness in reducing planning errors (Appendix D). Our model, as described in Section 4.2 and shown in Algorithm 1, selects K steps based on ranked verification scores. Higher verification scores suggest more accurate planning, making them a better measure than existing heuristic methods.
>
>
>
>
>
> > W3.1: missing-ablation: The hyper-parameters are not evaluated
>
> **A:** Per your request, the results are as follows:
>
> K=3: 60.85; K=4: 61.49
>
> P=1: 60.98; P=2: 61.49
>
>
> > W3.2: missing-ablation: vanilla ensemble LLM.
>
>
> **A:** Indeed, we've compared an LLM-enhanced tree-search, as shown in Table 1, Line 4. This method replaces Algorithm 1's sorting, ranking, and selecting steps with LLM, which can serve as a stronger baseline.
>
>
> > W4.1: confusing-presentation: The names of different variants of the EXOVIP model are not straightforward to know what the variant is. (TRS in section 5.1 and table 3)
>
> **A:** To clarify, in line 3, "ExoVIP w/o self-correctness" means using the TRS strategy only. In line 4, "ExoVIP w/o verification" means using the PSC only. Plans are to clarify this further.
>
>
> > W4.2: table 2 not match Table 1
>
>
> **A:** We separating our tree-search strategy and verification score for clear comparison. Our method uses a prediction score for tree-search, then employs an LLM to pick error-free planning traces, fixing the planning tree. This lets us assess the verification score alone. The verification score isn't part of our framework, hence its absence in Table 1. This matches the example in Appendix B with a value of 60.034, identical to Table 2's mixture score. We can offer more details if space allows.

---

> > ### Comment · Reviewer_MW2q · 2024-06-06
> >
> > Thanks for the rebuttal. I do not find the explanation for negative sampling convincing, and the presentation of the paper should be largely improved to make it easier to understand the technical details and results. Therefore, I kept my original rating.

---

> > ### Author Response · Authors · 2024-06-06
> > **Response to Reviewer MW2q**
> >
> > Thank you for your response. Could you please specify the reason you are not convinced? Negative sampling is a long-standing research topic, and in addition to the works we cite in our paper, there are many related studies such as negative sampling [1-5], adversarial sampling [6-9], and additional data-enhanced sampling in information retrieval [10-14]. One of the main designs of our framework is adopting negative sampling to normalize the confidence score for tree-based search across different tools and agents.  Our experiments also demonstrate its effectiveness.
> >
> > Regarding the presentation, we have carefully addressed your concerns. Could you please specify any remaining points of confusion? Your suggestions are invaluable for us to further improve our work.
> >
> > [1] Hard Negative Mixing for Contrastive Learning.
> >
> > [2] Bundle Recommendation with Graph Convolutional Networks. SIGIR(2020)
> >
> > [3] Supervised Contrastive Learning. NIPS(2020)
> >
> > [4] Hard-Negatives or Non-Negatives? A Hard-Negative Selection Strategy for Cross-Modal Retrieval Using the Improved Marginal Ranking Loss. ICCV(2021)
> >
> > [5] Neighborhood Contrastive Learning for Scientific Document Representations with Citation Embeddings. EMNLP(2022)
> >
> > [6] Synthesizing Adversarial Negative Responses for Robust Response Ranking and Evaluation.
> >
> > [7] Adversarial training regularization for negative sampling based network embedding. Information Sciences(2021)
> >
> > [8] Adversarial Caching Training: Unsupervised Inductive Network Representation Learning on Large-Scale Graphs. TNNLS(2021)
> >
> > [9] Instance-wise Hard Negative Example Generation for Contrastive Learning in Unpaired Image-to-Image Translation. ICCV(2021)
> >
> > [10] Leveraging Social Connections to Improve Personalized Ranking for Collaborative Filtering. CIKM(2014)
> >
> > [11] Social Recommendation with Strong and Weak Ties. CIKM(2016)
> >
> > [12] Joint Geo-Spatial Preference and Pairwise Ranking for Point-of-Interest Recommendation. ICTAI(2017)
> >
> > [13] A Personalised Ranking Framework with Multiple Sampling Criteria for Venue Recommendation. CIKM(2017)
> >
> > [14] Socially-Aware Self-Supervised Tri-Training for Recommendation.

---

### Official Review · Reviewer_ouxC · 2024-05-10

**Rating:** 7
**Confidence:** 4
**Ethics Flag:** 1

**Summary:**

The work aimed to tackle the planning mistakes and execution errors from existing LLMs.
The proposed method, ExoViP, corrects errors at both the planning and execution stages through introspective verification.
ExoViP employs a verification module with three sub-verifiers to validate predictions and refine the reasoning trace planned by LLMs.
Experimental results show consistent improvements on five compositional reasoning tasks on standard benchmarks.
ExoViP has the potential to improve performance and generalization on open-domain multimodal challenges.

**Questions To Authors:**

Overall, it is a decent paper. Please address the weakness above.

**Reasons To Accept:**

The introduced "plug-and-play" EXOVIP for correcting errors is straightforward, and ablation studies demonstrate the effectiveness of the components.
The verification module uses a mixture of three sub-verifiers to validate predictions, providing a comprehensive and robust approach to error correction.
Experimental results show consistent improvements on five compositional reasoning tasks, demonstrating the effectiveness of the ExoViP method.

**Reasons To Reject:**

There are not many serious weaknesses in the paper.
1. Things to improve include the fact that Figure 1 is not very comprehensive; some of the terms, e.g., LOC, VQA[BOX], are unexplained in the beginning. Also, the writing in section 4 should link to Figure 1 to explain each component regarding its target in the pipeline. It would be better to include a high-level figure at the beginning.
2. Another weakness is that the proposed verification is simply a combination of existing works. Also, the idea is in line with the motivation of the Tree of Thoughts, where performance gain is expected.
3. Some typos need to be fixed, e.g., Appendix ?? at section 5.3

---

> ### Author Rebuttal · Authors · 2024-05-31
>
> Thank you for your constructive comments and acknowledging of our effective method. We carefully address each of your concerns as follow.
>
>
> > W1: Things to improve include the fact that Figure 1 is not very comprehensive
>
> **A:** Thank you for your suggestion. We aim to create a clearer and more polished figure to effectively present our framework. Specifically, we will  include subverifiers and TRS in figure 1. For the high-level figure, we will extend figure2 to our framework overview. Due to the limited page numbers, we will add in the final version (with 1 extra page).
>
>
> > W1: Another weakness is that the proposed verification is simply a combination of existing works.
>
> **A:** Thank you for your feedback, to address your concerns directly and provide a more comprehensive understanding, I will delve deeper into the analysis of the verifiers and tree-based search algorithm with other works:
>
> - **Our verification strategy is more than introduce additional modules:**
>     - **Significance of Negative Sampling**: In Section 5.1.1, our findings indicate that verifiers on their own may perform suboptimally. This observation led us to implement a negative sampling strategy. This score is not an absolute measure but a relative one, designed to robustly reflect the model's confidence in its predictions.
>     - **Synergy of Multiple Subverifiers**: We never introduce new modules from outside, all the verifiers comes from the operation modules. And Table 2 highlights the significance of combining different subverifiers.
>
> - **Our tree-based strategy v.s ToT, MCTS**
>     - **Interplay Between Verifiers and Reasoning Trace**: A key innovation of ours is the integration of verification scores to optimize the reasoning trace. As shown in Figure 3, when we combine the tree-based search algorithm with verification scores, there is an improvement in the variance of these scores. This enhancement is instrumental in differentiating between various reasoning traces, ultimately aiding in the identification and selection of the most accurate one. Thus, the subverifiers and the trace search algorithm mutually reinforce each other's effectiveness.
>     - **Efficiency:** As we stated in Appendix E, we assert that our beam searchbased method can achieve an better equilibrium between efficiency and effectiveness comparing with the previous search strategy.
>
> > W1: Some typos need to be fixed.
>
> **A:** Thank you for your suggestions, we would like to revise our paper carefully.

---

> > ### Comment · Reviewer_ouxC · 2024-06-06
> >
> > I have read through the reviews and rebuttals.
> > Most of my concerns are addressed.
> > I'll keep my rating.

---

### Official Review · Reviewer_cnU5 · 2024-05-11

**Rating:** 6
**Confidence:** 3
**Ethics Flag:** 1

**Summary:**

This paper introduces a new plug-and-play approach called 'EXOVIP,' to improve compositional visual reasoning and address two challenges in current visual programming: module prediction error and planning error.
"Exoskleton" verification modules (that consists in a mix of image-text matching verifier, image captioning verifier and VQA verifier) are used to verify the prediction after every step, followed by corrections or refinings.
The experiments conver six compositional visual reasoning tasks covering real and abstract images/videos, and shows consistent improvement.

**Questions To Authors:**

1. Could you add a short definition of the prediction error and planning error or a reference to the definitions in the caption of Figure 1, so that the very first figure is more self-contained & easy to understand?

2. It could be clearer for the reader if the authors explicitly specify "NLVR2" in the caption of Table 4, as specified in Section 5.3.

3. Typo in 5.3, paragraph "Text-guided Image Editing": "Appendix ??"

Not a question, but I appreciate the analysis about errors/efficiency conducted in the Appendix.

**Reasons To Accept:**

The writing is pretty clear and easy to follow.

The paper proposes a novel and clear approach using verification modules to deal with the planning and module errors in compositional visual reasoning.

The paper empirically evaluated the approach on a variety of tasks and shows consistent improvement across the tasks, notable for some (e.g. for the compositional VQA tasks).
The evaluations cover both real images and abstract synthetic ones.

**Reasons To Reject:**

There's a concern about the efficiency of the method, as Table 9 shows that the total inference time of EXOVIP is 9.6x the time of LLaVA-1.5-7B. The authors indicated that Internet latency may be a cause, it could be helpful to have an estimate using open-source LLMs.

There's inconsistencies in the paper about the number of tasks and it's quite confusing: sometimes the paper refers to "the outcome has been a uniform enhancement in performance across six diverse tasks", whereas at multiple other places (including in the abstract and appendix) the paper refers to "consistent improvements on five compositional reasoning tasks".

---

> ### Author Rebuttal · Authors · 2024-05-31
>
> Thank you for your recognition of our methods' potential to improve performance and generalization on open-domain multimodal challenges. We carefully address each of your concerns as follow.
>
>
> > W1: There's a concern about the efficiency of the method, as Table 9 shows that the total inference time of EXOVIP is 9.6x the time of LLaVA-1.5-7B. The authors indicated that Internet latency may be a cause, it could be helpful to have an estimate using open-source LLMs.
>
> **A:** We have replaced GPT-3.5-turbo to Llama-2-13B-AWQ. Specifically, we have deployed the original Llama-2-13B-AWQ model on a single Nvidia A800 GPU using the vllm-4.0 framework on a cluster, without employing any additional acceleration strategies. The following results were obtained:
>
> | model   | Total Inference Time (s) |  Planning Time (s)   | Module Inference Time (s) |
> | ------- | -------------------- | --- | -------------------------- |
> | ExoViP(GPT-3.5-turbo)  | 4.32                 |   3.64  | 0.68         |
> | ExoViP(Llama-2-13B-AWQ)  | 3.94                 |   2.99  | 0.95         |
>
> We observe that the planning time decreased but module time increased. We believe this is attributed to the fact that the LLaMA model induces more planning errors, resulting in a significant rise in search steps. In the future, we aim to implement more sophisticated models, employing GPU acceleration tactics.
>
> > W2: There's inconsistencies in the paper about the number of tasks and it's quite confusing.
>
> **A:** Thank you the catch. This is due to the new task we performed close to the deadline. The total number of tasks is six. We apologize for the confusion. We will carefully revise our presentation accordingly.
>
>
> > Q1: Could you add a short definition of the prediction error and planning error or a reference to the definitions in the caption of Figure 1, so that the very first figure is more self-contained & easy to understand?
>
> **A:** Thank you for your suggestion. We will add a brief definition and references of "planning error" and "module error" as stated in the Introduction Section to Figure 1.
>
>
> > Q2: It could be clearer for the reader if the authors explicitly specify "NLVR2" in the caption of Table 4, as specified in Section 5.3.
>
> **A:** Thank you for your suggestion; we will change "NLVR" to "NLVR2" accordingly.
>
> > Q3: Typo in 5.3, paragraph "Text-guided Image Editing": "Appendix ??"
>
> **A:** Thanks. We have fixed these typos and will carefully proofread our paper.

---

> > ### Comment · Reviewer_cnU5 · 2024-06-05
> > **Response to the authors**
> >
> > Thanks to the authors for the responses and clarification. I will keep my current score for now, given that the potential gains seem to be mitigated by the overhead in inference time.

---

> > > ### Author Response · Authors · 2024-06-06
> > >
> > > Thank you for your useful comments and responses. As identified in our introduction, our proposed framework is generalizable and training-free, thus we believe such general adaptation capability can mitigate the minor inference overhead as indicated in response to **W1**. Moreover, since our LLMs are mainly used for tool-use, some lightweight tool-based LLMs can further reduce the inference overhead and strengthen the performance gain.

---

### Official Review · Reviewer_pE53 · 2024-05-23

**Rating:** 6
**Confidence:** 3
**Ethics Flag:** 1

**Summary:**

The paper presents an "exoskeleton" approach to compositional visual reasoning as an amelioration to the visual programming work. Considering the planning error and the module execution error in the visual reasoning process, this work introduces three verifiers and a series of other mechanisms (TRS, self-correction) to enhance the performance in GQA and many other reasoning tasks.

**Questions To Authors:**

See the weaknesses above.

**Reasons To Accept:**

+ The work introduces verifiers to rectify the errors in the reasoning process, which sounds reasonable.
+ The writing is clear and easy to follow.
+ It involves extensive experiments spanning multiple vision benchmarks.

**Reasons To Reject:**

+ The introduction of 3 verifiers, together with TRS and self-correction is supposed to cause extra latency at inference. So I am skeptical about whether the gains in performance are worth it.
+ The author(s) claim that the proposed exovip framework can mitigate the planning error and present an error analysis in Appendix, which is informative. However, I suggest that presenting the accuracy of planning as an important intermediate step would be more intuitive.
+  The introduction of 3 verifiers lacks a clear motivation and we have little idea why they should be designed in this way. Maybe a more in-depth analysis of how the verifiers collaborate with each other would be helpful.

---

> ### Author Rebuttal · Authors · 2024-05-31
>
> Thank you for your comments and acknowledgment of our reasonable idea.
>
> > W1: The introduction of 3 verifiers, together with TRS and self-correction is supposed to cause extra latency at inference. So I am skeptical about whether the gains in performance are worth it.
>
> **A:** In Appendix E, we present an analysis of inference time, as detailed in Table 9. The data indicates that the majority of the cost arises from planning time, which we attribute to the OpenAI API. To mitigate this latency, we plan to deploy a LLM locally. Furthermore, the current performance gains are constrained by our choice of verifier. For the sake of a fair comparison, we have selected the verifier from the operation modules. Future improvements could be achieved by introducing stronger verifiers.
>
>
>
> > W2: The author(s) claim that the proposed exovip framework can mitigate the planning error and present an error analysis in Appendix, which is informative. However, I suggest that presenting the accuracy of planning as an important intermediate step would be more intuitive.
>
> **A:** We thank you for your constructive suggestion. To clarify our approach, we have emphasized the connection to Appendix D in our introduction and will incorporate additional details regarding the accuracy of planning in the main body of our text.
>
>
> > W3.1: The introduction of 3 verifiers lacks a clear motivation and we have little idea why they should be designed in this way.
>
> **A:**  Your query is valid. We have chosen VQA, Caption, and CLIP as verifiers for the following reasons:
>
> - We selected verifiers from the operation modules to ensure a fair comparison.
> - We opted for three verifiers to maintain a balance between computational efficiency and performance.
> - We chose three distinct types of Vision-and-Language (VL) models that are commonly employed in multitask learning[1].
>
> [1] Lu, Jiasen, et al. “12-in-1: Multi-Task Vision and Language Representation Learning.” 2020 IEEE/CVF Conference on Computer Vision and Pattern Recognition (CVPR), 2019, pp. 10434-10443.
>
> > W3.2 Maybe a more in-depth analysis of how the verifiers collaborate with each other would be helpful.
>
> **A:** We appreciate your recommendation. To provide further clarity, we describe the distinct roles of the verifiers as follows:
>
> - VQA: Offers a discriminative viewpoint for VL comprehension.
> - Caption: Generates results with a broader, open-vocabulary approach.
> - CLIP: Demonstrates alignment and representation learning within VL models.

---

### Decision · Program_Chairs · 2024-07-10

**Decision:**

Accept

**Comment:**

The paper introduces ExoViP, a "plug-and-play" method to correct errors in both the planning and execution stages of compositional visual reasoning through introspective verification. The proposed verification module, consisting of three sub-verifiers, validates predictions after each reasoning step and refines the reasoning trace planned by LLMs. The reviewers acknowledge the clear writing, the novelty of the approach, and the extensive experiments demonstrating consistent improvements across various compositional reasoning tasks. Although some concerns were raised regarding the efficiency of the method, the inconsistencies in the number of tasks mentioned, and the lack of clarity in certain aspects of the algorithm and hyperparameter evaluation, the overall consensus is that the paper makes a valuable contribution to the field. The AC believes that the negative sampling part is acceptable, given the explaination of the authors from the rebuttal. Considering the strengths and the potential for improved performance and generalization on open-domain multimodal challenges, I recommend accepting this paper.

[comment from the PCs] The term "antonym" has a very specific linguistic definition, which does not align with how it's used in this paper. This led to misunderstandings by the reviewers, which significantly overshadow the evaluation of the paper. What more, using this term does not help the paper, it just hurts it. For the next version of the paper, we ask the authors to replace this term with something more appropriate. This is critical to meet publication standards.